# Synthesis and Characterization of Hydrophobic and Low Surface Tension Polyurethane

**Autumn M. Rudlong and Julie M. Goddard ***

Department of Food Science, Cornell University, Ithaca, NY 14853, USA; amr452@cornell.edu
* Correspondence: goddard@cornell.edu; Tel.: +1-607-255-8622

**Abstract:** Polyurethane is a common polymeric coating, providing abrasion resistance, chemical durability, and flexibility to surfaces in the biomedical, marine, and food processing industries with great promise for future materials due to its tunable chemistry. There exists a large body of research focused on modifying polyurethane with additional functionalities, such as antimicrobial, non-fouling, anticorrosive action, or high heat resistance. However, there remains a need for the characterization and surface analysis of fluoro-modified polyurethanes synthesized with commercially available fluorinated polyol. In this work, we have synthesized traditional solvent-borne polyurethane, conventionally found in food processing facilities, boat hulls, and floor coatings, with polyurethane containing 1%, 2%, and 3% perfluoropolyether (PFPE). Polyurethane formation was confirmed by attenuated total reflectance Fourier-transform infrared (ATR-FTIR) spectroscopy, with the urethane band forming at 1730 cm$^{-1}$ and the absence of free isocyanate stretching from 2275–2250 cm$^{-1}$. X-ray photoelectron spectroscopy (XPS) was used to confirm perfluoropolyether polymerization with an increase in the atomic percentage of fluorine. Wettability and hydrophobicity were determined using a dynamic water contact angle with significant differences in advancing the water contact angle with the inclusion of perfluoropolyether blocks (PU–*co*–1PFPE 131.5° ± 8.0, PU–*co*–2PFPE 130.9° ± 5.8, and PU–*co*–3PFPE 128.8° ± 5.2) compared to the control polyurethane (93.6° ± 3.6). The surface orientation of fluorine supported the reduced critical surface tensions of polyurethane modified with PFPE (12.54 mN m$^{-1}$ for PU–*co*–3PFPE compared to 17.19 mN m$^{-1}$ for unmodified polyurethane). This work has demonstrated the tunable chemical qualities of polyurethane by presenting its ability to incorporate fluoropolymer surface characteristics, including low critical surface tension and high hydrophobicity.

**Keywords:** polyurethane; nonfouling coatings; fluorinated polymers; hydrophobic; low surface energy

## 1. Introduction

Polyurethane coatings are common in marine environments, hospitals, and food manufacturing industries to protect floors, equipment, and high-use objects from abrasion and exposure to cleaning and sanitizing agents. The chemical makeup of polyurethane consists of soft (long-chain polyols) and hard blocks (di- or triisocyanates), with the soft blocks providing elasticity and the hard blocks providing structure and abrasion resistance. While polyurethanes have been in use industrially for nearly 50 years, new technologies have evolved over the past two decades to introduce new functionalities to improve durability, surface characteristics, and chemical resistance/interaction [1–3]. For example, Hu et al. synthesized polyurethane adhesives with quaternary ammonium antibacterial properties [4]; Hwang et al. synthesized UV-curable fluorinated polycarbonate polyurethane dispersion [5]; and Hill et al. explored L-tyrosine-based polyurethanes and physiochemical interactions with endothelial cells [6]. An interesting area of research focuses on the modification of polyurethane to reduce overall surface energy and hydrophobicity. It has been shown that a relationship exists between surface energy and the advancing contact angle of a liquid, which Young's equation illustrates [7]. Improvements to the overall hydrophobicity of a

material can be made by the incorporation of known low-surface-energy copolymers, like fluoropolymers, with wide coating applications (e.g., antifog/anti-icing, antifouling, drag reducing, self-cleaning). Researchers developing fluorine-modified polymers have proven the polymers exhibit characteristics commonly associated with fluoropolymers (e.g., low surface energy (6–15 mN m$^{-1}$) [8], chemical resistance) yet maintain characteristics of the base polymer [9–12]. The hydroxyl and isocyanate reaction in polyurethanes opens the door for many interesting modifications incorporating fluorinated polyols [13–16], fluorinated chain extenders [17,18], and even fluorinated isocyanates [19]. Takakura et al. synthesized a polyurethane using fluorinated diisocyanate for blood compatible materials [19]; however, this approach is not often repeated. The most common approach to synthesize fluorinated polyurethanes is to use fluorinated polyols or diols due to a large atomic percentage of fluorine available to introduce hydrophobicity and low surface tension characteristics.

Prior work with perfluoropolyether (PFPE) in polyurethane has shown promising results, including Ajroldi and Tonelli, who reported on the synthesis of a PFPE diol and a polyurethane block copolymer that possessed the mechanical characteristics of polyurethane and the surface characteristics of a fluorinated material [13]. Additionally, Choi et al. previously reported the atom transfer radical polymerization of a triblock copolymer consisting of modified Fluorolink® E10-H diol (PFPE diol) and methacrylate that showed promising surface energy values (22–25 mN m$^{-1}$) and hydrophobicity [20]. Gu et al. reported the synthesis of a triblock polyurethane–poly (isobornyl methacrylate)–perfluoropolyether copolymer with Fluorolink® E10-H diol that increased the water contact angle to 140° [21]. These works offered important new findings in the development of fluorinated polyurethane and applications of commercially available PFPE. Nevertheless, there remains a need for facile polyaddition synthesis and detailed surface characterization, including analysis of the critical surface tension, using the Zisman Plot method, of polyurethane modified with commercially available perfluoropolyethers. In this study, three polyurethane–*co*–perfluoropolyether coatings with varying percentages of commercially available Fluorolink® E10-H diol were synthesized via polyaddition reaction with 4,4′-Methylenebis (phenyl isocyanate) and poly (propylene glycol) with the aim to incorporate fluoropolymer surface characteristics within polyurethane. Surface characteristics, including chemical composition, water contact angle, and critical surface tension using the Zisman Plot approach, were determined for this coating.

## 2. Materials and Methods

### 2.1. Materials

Fluorolink® E10-H (perfluoropolyether) was provided in viscous liquid form by Solvay (Bollate, MI, Italy). In addition, 4,4′-Methylenebis (phenyl isocyanate) 98% (MDI) solid flake, poly (propylene glycol) molecular weight 725 (PPG) viscous liquid, anhydrous ethylene glycol (EG), anhydrous dimethylformamide 99.8% (DMF), tin (II) 2-ethylhexanoate, ethylene glycol, glycerol, and hexane were purchased from Millipore Sigma (Burlington, MA, USA). Type 304 2B finish stainless steel coating panels were purchased from Q-Lab Corporation (Cleveland, OH, USA). An adjustable-thickness 6-inch-width drawdown bar for coating was purchased from Paul N. Gardner Company (Pompano Beach, FL, USA).

### 2.2. Synthesis of Polyurethane and Polyurethane–Co–Perfluoropolyether

Polyurethane and polyurethane–*co*–perfluoropolyether were synthesized via a two-step prepolymer route using MDI as the hard segment and PPG or PPG and perfluoropolyether (PFPE) as the soft segment (Scheme 1, Table 1). Reactions took place in a 5-neck round-bottom reaction vessel equipped with an 80 °C silicon oil bath, an overhead stirrer, a temperature probe, a condenser with bubbler, an addition funnel, and an ultrapure nitrogen inlet. Safety precautions were taken in the storage and handling of MDI, as this compound possessed additional hazards. MDI was stored at −18 °C in an airtight and desiccated glass storage vessel and was allowed to reach 25 °C in a ventilated hood prior to removal from the storage vessel. All handling of MDI compounds was performed in a

ventilated hood, and proper personal protective equipment, including gloves, goggles, and an N95 respirator, were worn by the handler. For the control, polyurethane (PU) 6.90 mmol poly (propylene glycol) (PPG) was dissolved in 8 mL anhydrous DMF and placed into the nitrogen-purged reaction vessel. For polyurethane–*co*–perfluoropolyether (PU–*co*–PFPE) synthesis, 6.90 mmol PPG and 0.03 mmol (1 wt%), 0.06 mmol (2 wt%), or 0.09 mmol (3 wt%) perfluoropolyether (PFPE) were dissolved in 8 mL anhydrous DMF. An additional 8 mL anhydrous DMF was used to rinse the beaker of remaining reagent for all preparations. Tin (II) 2-ethylhexanoate was added at 0.5 wt% to catalyze the reaction [20]. MDI was added at a consistency near 1:1 polyol:isocyanate molar ratio for each treatment, thus increasing with increasing wt% PFPE. For the control PU, 10.27 mmol MDI was added, followed by stirring at 500 rpm for 30 min. For 1%, 2%, and 3% PFPE coatings, 10.27 mmol, 10.31 mmol, and 10.35 mmol MDI, respectively, were added, followed by stirring at 500 rpm for 30 min to form the prepolymer. After 30 min, 4.03 mmol ethylene glycol (EG) chain extender dissolved in 4 mL anhydrous DMF and an additional rinse of 4 mL anhydrous DMF were added dropwise to the prepolymer mixture over 10 min using an addition funnel. After the chain extender addition, residual isocyanate reacted for an additional 2.5 h at 80 °C at 500 rpm. After completion of polymerization, the polymer was extracted from the vessel and vacuum degassed at $10^{-2}$ mBar at 20 °C for 1 h.

**Scheme 1.** Two-step prepolymer reaction of MDI (**A**), PPG (**B**), and PFPE (**C**) forming the prepolymer (**D**). Chain extender (**E**) added to form final PU–*co*–PFPE polymer (**F**).

**Table 1.** Formulation of polyurethane and polyurethane–*co*–perfluoropolyethers.

|  | MDI (mmol) | PPG (mmol) | EG (mmol) | PFPE (mmol) |
|---|---|---|---|---|
| Polyurethane | 10.27 | 6.90 | 4.03 | - |
| PU–*co*–1PFPE | 10.27 | 6.90 | 4.03 | 0.03 |
| PU–*co*–2PFPE | 10.31 | 6.90 | 4.03 | 0.06 |
| PU–*co*–3PFPE | 10.35 | 6.90 | 4.03 | 0.09 |

### 2.3. Coating and Curing

Type 304 2B finish stainless steel coupons were used as solid support for coatings. Polymer coatings were applied with a drawdown bar to 0.025-inch thickness. Additionally, polymer was poured into custom-fabricated polytetrafluoroethylene (PTFE) molds mea-

suring 1 cm by 1 cm. PTFE molds were prepared by washing with 1% anionic detergent solution and rinsed with deionized water followed by oven drying at 60 °C. The coatings were cured at 80 °C with full vacuum ($10^{-2}$ mBar) for 24 h. Following curing, the coatings were cooled to room temperature before characterization.

### 2.4. Instrumental

The numbers for average molecular weight (Mn), weight average molecular weight (Mw), and dispersity (Mw/Mn) were collected using a Waters Ambient-Temperature GPC (Waters Corporation, Milford, MA, USA) equipped with triple-detection capability: a Waters 410 differential refractive index detector; a Waters 486 UV–Vis detector; and a Wyatt Technologies TREOS three-angle light-scattering detector. Samples were dissolved in tetrahydrofuran (THF). Urethane formation was confirmed using attenuated total reflectance Fourier-transform infrared (ATR-FTIR) spectroscopy. Spectra were collected using an IRTracer-100 FTIR spectrometer (Shimadzu Scientific Instruments, Inc., Kyoto, Japan) equipped with a diamond ATR crystal. Spectra were collected using Happ–Genzel apodization (4 cm$^{-1}$, 32 scans). Samples were analyzed using a Scienta Omicron ESCA-2SR (Scienta Omicron GmbH, Taunusstein, Germany) with operating pressure ca. $1 \times 10^{-9}$ mBar. Monochromatic Al K$\alpha$ X-rays (1486.6 eV) were generated at 300 W (15 kV; 20 mA). The analysis spot size was 2 mm in diameter with a 0° photoemission angle and a source-to-analyzer angle of 54.7°. A hemispherical analyzer determined electron kinetic energy, using a pass energy of 200 eV for wide/survey scans and 50 eV for high resolution scans. All samples were charge neutralized using a low-energy electron flood gun. 1H NMR spectra of 25 mg mL$^{-1}$ in dimethyl sulfoxide-d6 PU and PU–$c$–PFPE were collected by a Bruker AV500 (Bruker, Billerica, MA, USA). Spectra were analyzed using MestreNova MNova.

### 2.5. Surface Hydrophobicity and Critical Surface Tension

Surface hydrophobicity and surface tension were measured using dynamic contact angles on an Attension Theta Optical Tensiometer (Biolin Scientific, Stockholm, Sweden). Briefly, advancing water contact angles were determined by depositing a small droplet of Milli-Q purified water onto the surface of each sample and placing the needle in the droplet and expanding the droplet at a rate of 0.5 μL s$^{-1}$. The advancing angle was defined as the maximum angle prior to an advance in droplet baseline [21]. Images of droplets were recorded at a rate of 14 frames s$^{-1}$, and the angles were measured using the Young–Laplace method. The receding water contact angle was determined by subsequently withdrawing a water droplet from the surface at a rate of 0.5 μL s$^{-1}$. Receding angles were defined as the minimum angle prior to the recession of the droplet baseline [22,23]. The surface tension of the materials was calculated using the Zisman Plot method [24,25]. Briefly, four liquids with known surface tensions were individually deposited onto the material surface at a rate of 0.5 μL s$^{-1}$, and the advancing contact angle was taken at the maximum angle prior to the advance in droplet baseline. The cosine of each contact angle (θ) was calculated, and averages were plotted against the known surface tension of the liquids. A non-linear quadratic model was fitted to the data, and a replicates test was performed to determine if the chosen model was adequate. Liquids used to determine the surface tension of PU and PU–$co$–PFPE included Milli-Q purified water, ethylene glycol, glycerol, and hexane.

### 2.6. Statistics

ATR-FTIR scans were taken from four coupons from two separately synthesized batches for a total of eight scans, with characteristic band analysis performed on OriginPro 2021 (OriginLab Corporation, Northampton, MA, USA) and KnowItAll Software (BioRad Laboratories, Hercules, CA, USA). Spectra displayed were randomly chosen by assigning numbers (1–8) and using a random number generator to select the spectra. ATR-FTIR data not displayed in this article are available by request. Surface hydrophobicity and surface tension measurements were performed on four individual coupons from two

separately synthesized batches and analyzed using one-way analysis of variance (ANOVA) ($p < 0.05$) with Tukey's HSD multiple comparisons ($p < 0.05$) in addition to the line fitting and replicates test using GraphPad Prism version 7.05 (GraphPad Software, San Diego, CA, USA). For surface hydrophobicity, technical replicates ($n = 4$) were averaged, and the experiment was repeated, providing two independent averages that were used for ANOVA analysis.

## 3. Results and Discussion

### 3.1. Synthesis

PFPE-modified polyurethanes were synthesized via polyaddition polymerization in which prepolymer (Scheme 1D) was formed by the reaction between diisocyanate MDI (Scheme 1A) and the mixture of two polyols, PPG (Scheme 1B) and PFPE (Scheme 1C). The control polyurethane followed an identical synthesis route; however, PPG would replace PFPE in prepolymer D. Following the formation of the prepolymer, ethylene glycol (Scheme 1E) reacts with terminal isocyanate groups to further extend the polymer chains (Scheme 1F). Control and PFPE-modified polyurethanes were poured into untreated PTFE molds to create a free-standing polyurethane film for further characterization. Figure 1 depicts free-standing polyurethane coupons. Visual observations of the coatings after demolding showed retained transparency in coatings containing PFPE compared to unmodified polyurethane. During preliminary experimentation, 5 wt% PFPE polyurethane was synthesized but did not retain transparency and could not be removed from the mold; thus, 3 wt% PFPE polyurethane was the highest PFPE content explored. While nanoscale imaging (with roughness determination) is essential for ultrahydrophobic coatings that manipulate nanoscale topography to achieve hydrophobicity, our approach does not include such nanotopographical modifications. Rather, our approach manipulates the chemistry of the polyurethane coating to introduce fluorinated groups that are responsible for the enhanced hydrophobicity. For these reasons, and in line with other publications on chemically (not topographically) modified polyurethane coatings, our analysis includes macroscale imagery (Figure 1) and chemical analyses (in the following sections).

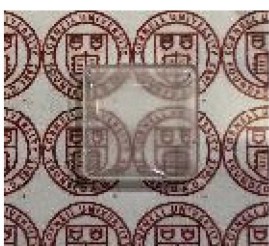 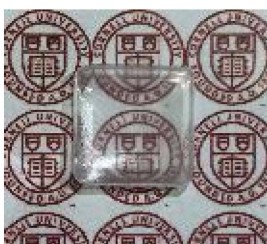 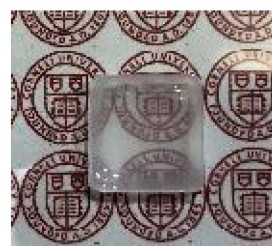 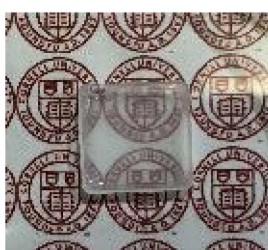

**Figure 1.** Coatings, 1 cm by 1 cm, cured and removed from PTFE molds. From left to right: control polyurethane, PU–*co*–1PFPE, PU–*co*–2PFPE, and PU–*co*–3PFPE. Coupons were collected from the PTFE mold, and those with bubbles or missing edges were discarded. The samples displayed in this figure reflect characteristics of the larger batch, including size, shape, intact edges, no bubbles present, and visual clarity.

Polymerization of polyurethane and polyurethane–*co*–perfluoropolyether proceeded for three hours and produced high molecular weight polymers. The number and weight average molecular weights for control polyurethane and PFPE polyurethanes were determined using gel permeation chromatography in THF (Table 2). PFPE-modified polyurethane retained desirable high $M_n$ and $M_w$ exhibited by control PU with $M_n$ of 26,000, 28,900, 29,200, and 26,400 Daltons for control polyurethane, PU–*co*–1PFPE, PU–*co*–2PFPE, and PU–*co*–3PFPE, respectively. The increase in molecular weight is due to increased chain length and chain entanglement within the polymers, which create a material with better mechanical stability and durability. The polydispersity indices (PDIs) determined are greater than one and less than two, indicative of a well-controlled step-growth polymerization, and are consistent with literature reporting the synthesis of polyurethane [20].

**Table 2.** Number average molecular weight ($M_n$), weight average molecular weight ($M_w$), and polydispersity for polyurethane, PU–*c*–1PFPE, PU–*c*–2PFPE, and PU–*c*–3PFPE.

|  | $M_n$ (Daltons) | $M_w$ (Daltons) | Polydispersity |
|---|---|---|---|
| Polyurethane | 26,000 | 41,890 | 1.61 |
| PU–*co*–1PFPE | 28,900 | 45,912 | 1.59 |
| PU–*co*–2PFPE | 29,200 | 49,341 | 1.69 |
| PU–*co*–3PFPE | 26,400 | 43,628 | 1.65 |

*3.2. Characterization*

Attenuated total reflectance Fourier-transform infrared spectroscopy (ATR-FTIR) was utilized to determine the chemical structure of the synthesized polymers. Polyurethane, PU–*co*–1PFPE, PU–*co*–2PFPE, and PU–*co*–3PFPE presented absorbance bands at 1525 cm$^{-1}$, indicative of urethane linkage formed between the isocyanate and the hydroxyl groups (Figure 2B). In addition, all samples displayed absorbance bands located at 1725 cm$^{-1}$, which indicated the presence of ester groups found in the reaction between isocyanate and hydroxyl end groups of the perfluoropolyether and ethylene glycol chain extender. This band was expected in all polymers, including the control polyurethane, due to the reaction with excess isocyanates and the chain extender. Polyurethane, PU–*co*–1PFPE, PU–*co*–2PFPE, and PU–*co*–3PFPE presented absorbance bands at 1600 cm$^{-1}$, indicating the cyclic carbons found in methylenebis (phenyl isocyanate). However, increases in band absorbance at 1203 cm$^{-1}$ were seen in polyurethanes containing PFPE compared to the control PU (Figure 2C). Peak deconvolution was also used to further analyze the region where a fluorocarbon band could be found (1500–900 cm$^{-1}$) (Supplementary Materials Figures S1–S8). Further characterization using $^1$H NMR indicated resonance centered at 1.00 ppm, corresponding to the methyl group found in polypropylene glycol (Figure 3A,B). For the polyurethane samples, resonance centered at 1.2 ppm can be attributed to the methyl groups of the polypropylene glycol. The perfluoropolyether (PFPE) (Figure 3A) displayed resonance centered at 3.3 ppm, which corresponds to the ethyl ether repeat units. The absolute integration for ethyl ether was calculated to be 9774.23, compared to 89,767.12, 76,421.28, 79,782.57, and 80,600.88 for control polyurethane, PU–*co*–1PFPE, PU–*co*–2PFPE, and PU–*co*–3PFPE, respectively (Supplementary Materials Table S1). Interestingly, the absolute area drops with the addition of PFPE and rises with the increasing amount of PFPE added. This can be attributed to the increasing amount of ethyl ether groups in the PFPE repeat units. Again at 3.3 ppm, the control polyurethane displayed resonance, which can be attributed to the polypropylene glycol and ethylene glycol. Given this overlap in resonance, PU–*co*–3PFPE possesses both signals.

The surface chemistries of PU and PFPE-modified PU were determined by XPS surface analysis. Survey scans indicated the successful incorporation of perfluoropolyether into the polymer coatings by the increase in fluorine atomic percentage 0.82% in polyurethane, 18.35% in PU–*co*–1PFPE, 20.67% PU–*co*–2PFPE, and 21.57% in PU–*co*–3PFPE (at. %) (Table 3). Increasing surface fluorine at. % with increasing PFPE concentration indicate the fluorine groups were successfully bound within the PU structure. Further, because XPS is a very surface-sensitive analytical technique, probing only 1–3 nm into the material, the increasing fluorine content of these top several nanometers suggests that the fluorine groups are surface oriented, an important feature for lowering surface tensions of materials. Atomic percentages of nitrogen also increased as more MDI was added to react with excess hydroxyl groups of the additional polyol from perfluoropolyether. Atomic percentages of carbon decreased with the addition of PFPE, as expected due to the increased ratio of fluorine within PFPE polymer segments and the decreased ratio of carbon within PPG polymer segments.

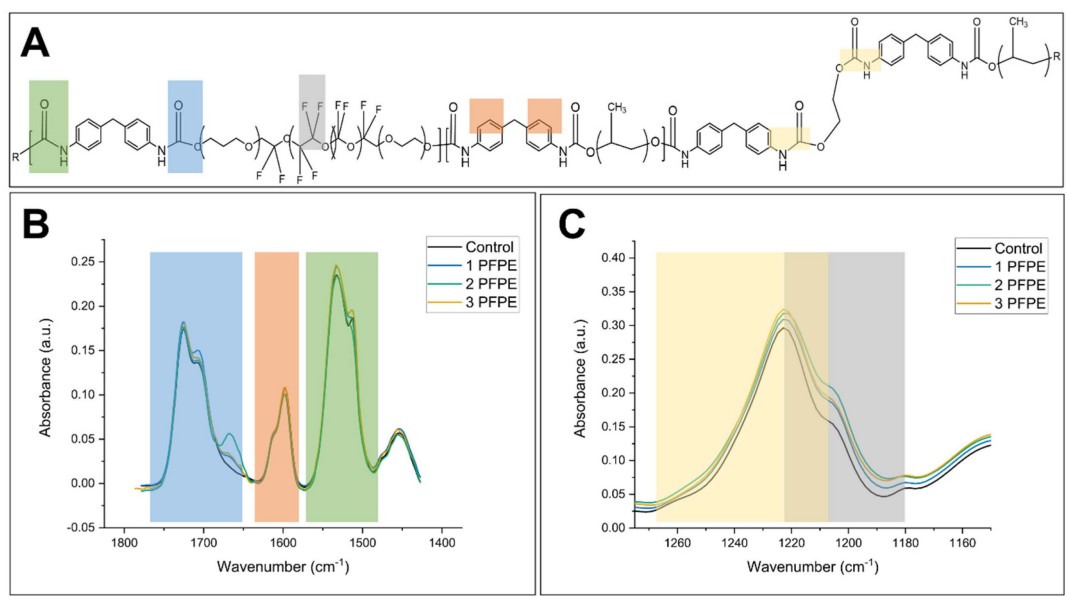

**Figure 2.** ATR-FTIR spectra of control polyurethane, PU–*co*–1PFPE, PU–*co*–2PFPE, and PU–*co*–3PFPE. (**A**) Spectra with blue highlighting the ester stretch region, green highlighting NH bend region, and gray highlighting the fluorocarbon stretch region. (**B**) The urethane region with blue highlighting the ester stretch, orange highlighting the cyclic alkyl stretch, and green highlighting the CHN bend. (**C**) Yellow highlighting CN stretching and the fluorocarbon region with gray highlighting the $CF_2$ band. Four individual coupons from two separately synthesized batches were scanned (*n* = 8). Spectra displayed were randomly chosen using a random number generator, and additional spectra are available upon request.

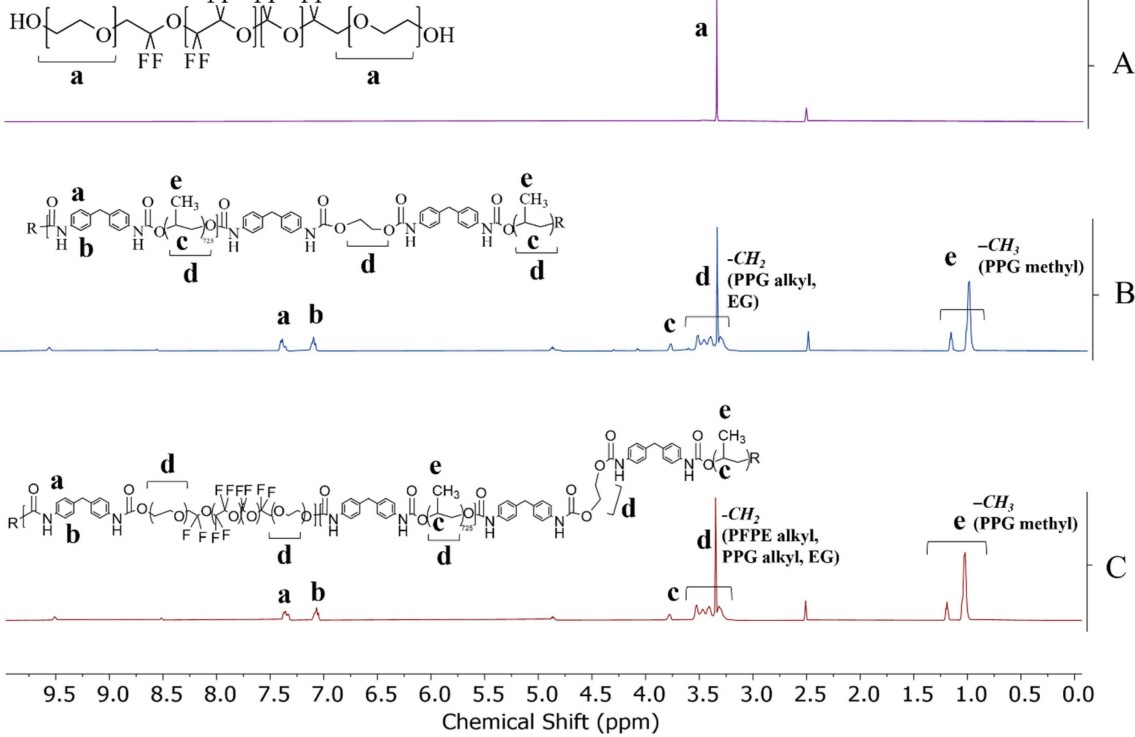

**Figure 3.** $^1$H NMR spectra of perfluoropolyether (**A**), polyurethane (**B**), and PU–*c*–3PFPE (**C**). Spectra collected in DMSO-$d_6$ (500 MHz).

**Table 3.** Summary of XPS spectral data including atomic percentage (at. %) utilizing spectral scans of polyurethane, PU–*co*–1PFPE, PU–*co*–2PFPE, and PU–*co*–3PFPE.

|  | Carbon (at. %) | Oxygen (at. %) | Nitrogen (at. %) | Fluorine (at. %) |
|---|---|---|---|---|
| Polyurethane | 73.52 | 23.94 | 1.73 | 0.82 |
| PU–*co*–1PFPE | 59.08 | 20.11 | 2.46 | 18.35 |
| PU–*co*–2PFPE | 58.19 | 20.67 | 2.84 | 20.67 |
| PU–*co*–PFPE | 58.75 | 17.36 | 2.32 | 21.57 |

Comparison and deconvolution of the high-sensitivity bonding of carbon between control polyurethane, PU–*co*–1PFPE, PU–*co*–2PFPE, and PU–*co*–3PFPE further indicated the polymerization of perfluoropolyether into the polyurethane backbone (Table 4). Surface contamination of PU–*co*–3PFPE at 280 eV led to differences in C-C and C-O bonding compared to PU–*co*–1PFPE and PU–*co*–2PFPE. Bonding associated with C=O increased with the addition of PFPE, as expected due to the larger amount of urethane linkages formed in PU–*co*–1PFPE, PU–*co*–2PFPE, and PU–*co*–3PFPE respectively.

**Table 4.** Summary of XPS high sensitivity carbon bonding indicating alkyl (C–C), ether (C–O–C), ester (O–C=O), carbonyl (C=O), and difluoromethylene (C–$F_2$) linkages of polyurethane, PU–*co*–1PFPE, PU–*co*–2PFPE, and PU–*co*–3PFPE.

|  | C–C (at. %) | C–O (at. %) | C=O (at. %) | C–$F_2$ (at. %) |
|---|---|---|---|---|
| Polyurethane | 73.33 | 23.87 | 2.80 | - |
| PU–*co*–1PFPE | 57.48 | 27.29 | 4.20 | 11.02 |
| PU–*co*–2PFPE | 54.57 | 24.19 | 6.16 | 14.41 |
| PU–*co*–3PFPE | 49.53 | 17.81 | 6.65 | 13.96 |

The surface hydrophobicity of stainless steel, control PU, and PU–*co*–PFPE was determined using a dynamic water contact angle with needle insertion. Here, we report both advancing and receding water contact angles as well as the calculated hysteresis values (the difference between advancing and receding water contact angles), to provide full information about the surface wettability of our coatings. Coatings containing perfluoropolyether had a significantly higher advancing water contact angle compared to control polyurethane and stainless steel (Table 5). Additionally, incorporation of perfluoropolyether created a hydrophobic ($\theta_a > 90°$) surface with PU–*co*–1PFPE having an advancing contact angle of 131.5° $\pm$ 8.0, PU–*co*–2PFPE of 130.9° $\pm$ 5.8, and PU–*co*–3PFPE of 128.8° $\pm$ 5.2. Control polyurethane was significantly different compared to stainless steel and PFPE polyurethane, and all PFPE polyurethane coatings were not significantly different from one another. Gu et al. reported a contact angle of 118° for synthesized fluorinated polyurethanes with 2.6 wt% Fluorolink® E10-H diol [21], and the 3 wt% PFPE polyurethane synthesized herein offers an improved water contact angle of 128.8°. There were no significant differences in the receding contact angles of the polyurethane coatings, but there was a significant difference in the receding contact angle of stainless steel compared to the coatings. The contact angle hysteresis was high where un-modified stainless steel had a hysteresis of 54.9°, polyurethane of 59.2°, PU–*co*–1PFPE of 104.3°, PU–*co*–2PFPE of 102.9°, and PU–*co*–3PFPE of 99.6°. In this study, our enhanced hydrophobicity correlates with increased hysteresis because the receding contact angles for all polyurethane variants remain the same. Santos et al. found similar contact angle hysteresis with unmodified stainless steel (78°) and stainless steel modified with polytetrafluoroethylene (70°) [26]. The reported high contact angle hysteresis could be attributed to the impact the fluorinated polymers have on water droplet movement. Yuan and Lee report that high contact angle hysteresis can be attributed to hydrophobic surface domains restricting the contracting movement of the receding droplet [27].

**Table 5.** Dynamic water contact angle for control polyurethane and fluorinated polyurethane. Significant differences between means are indicated by capital letters (Tukey's HSD, $p \leq 0.05$). Values represent means and standard deviations of four coupons from two independently synthesized batches ($n = 8$).

| | Advancing Contact Angle ($\theta_a$) | Receding Contact Angle ($\theta_r$) | Hysteresis |
|---|---|---|---|
| Stainless Steel | $73.4 \pm 17.8°$ [A] | $18.5 \pm 9.8°$ [A] | $54.9°$ |
| Polyurethane | $93.6 \pm 3.6°$ [B] | $34.4 \pm 4.3°$ [B] | $59.2°$ |
| PU–*co*–1PFPE | $131.5 \pm 8.0°$ [C] | $27.2 \pm 7.1°$ [AB] | $104.3°$ |
| PU–*co*–2PFPE | $130.9 \pm 5.8°$ [C] | $28.0 \pm 5.2°$ [AB] | $102.9°$ |
| PU–*co*–3PFPE | $128.8 \pm 5.2°$ [C] | $29.2 \pm 5.9°$ [B] | $99.6°$ |

The surface tension of the material is defined as that of a liquid which is just able to spread across the material surface ($\theta = 0$, $\cos\theta = 1$) [24,28]. The critical surface tension of the synthesized coatings and stainless steel were determined using the Zisman Plot method. The Zisman Plot method is best suited for low-energy and non-polar surfaces. This method utilizes three to five liquids with known surface tensions and plots the cosine of the advancing contact angle of each liquid against the known surface tensions of the selected liquids followed by extrapolation from a line of best fit. While linear fitting is most common for extrapolating the critical surface tension, many fluoropolymers are better suited for a non-linear model. Fox and Zisman explored the spreading of liquids on tetrafluoroethylene polymers and determined the fits to be parabolic rather than linear [29]. Therefore, a non-linear second order polynomial (quadradic) model was used in this analysis, and the equation of the line determined the critical surface tension when cos $\theta$ is equal to 1. To confirm the appropriate non-linear model was used for analysis, the replicates test was performed, and the goodness of fit is reported in Table 6. Additionally, the Zisman plot is available in Supplementary Materials (Figure S9). If a linear model were employed, the critical surface tension values for the modified polyurethanes would have been around 6 mN m$^{-1}$, which is not comparable to previously researched coatings, and the replicates test would report the model as inadequate. In this study, water (72.8 mN m$^{-1}$), glycerol (63.4 mN m$^{-1}$), ethylene glycol (47.7 mN m$^{-1}$), and hexane (18.4 mN m$^{-1}$) were used. The critical surface tensions of synthesized polyurethanes and stainless steel are presented in Table 6. The critical surface tension of polyurethane decreased with the inclusion of perfluoropolyether as control polyurethane possessed a critical surface tension of 17.19, PU–*co*–1PFPE of 12.11, PU–*co*–2PFPE of 12.25, and PU–*co*–3PFPE of 12.54 mN m$^{-1}$. However, increasing the amount of perfluoropolyether in the polyurethane did not alter the critical surface tension, as all PU containing PFPE had a critical surface tension of around 12 mN m$^{-1}$. Additional studies that determine the surface tension of stainless steel and modified polyurethane report values ranging from 28 to 43 mN m$^{-1}$ [26,30] for stainless steel and 10–47 mN m$^{-1}$ for modified polyurethanes [31,32]. As intriguing as these results are, it is difficult to compare the calculated critical surface tension of these synthesized materials to materials created by other researchers due to the large variation in methodology for determining surface tension. For example, Potschke et al. utilizes a pendant drop of the liquid polymer for characterizing the surface tension [31], and Erceg et al. utilized a sessile drop of only water and ethylene glycol and estimated the surface tension using the Owens–Wendt theory [32]. Though these methods vary, Erceg et al. reported surface tensions between 17 and 13 mN m$^{-1}$ for siloxane-modified polyurethanes. Interestingly, using a smaller weight percent (3 wt%) addition of PFPE displayed similar surface tensions (12.54 mN m$^{-1}$) compared to higher weight percent (30 wt%) of polydimethylsiloxane (13.23 mN m$^{-1}$) [32]. The low critical surface tensions and high water contact angles of the PU–*co*–PFPE further confirm the surface orientation of fluorine as found in XPS analysis; however, the atomic percentage of fluorine did not appear to influence these characteristics as the modified polymers possessed similar critical surface tensions.

**Table 6.** Critical surface tensions of stainless steel, control polyurethane, and fluorinated polyurethanes. Goodness of fit for linear regression of Zisman plot data. Values represent means from two replicates of independently synthesized coatings.

| | Critical Surface Tension $\gamma_{cr}$ (mN m$^{-1}$) | Goodness of Fit (r$^2$) |
|---|---|---|
| Stainless Steel | 17.49 | 0.99 |
| Polyurethane | 17.19 | 0.99 |
| PU–*co*–1PFPE | 12.11 | 0.99 |
| PU–*co*–2PFPE | 12.25 | 0.99 |
| PU–*co*–3PFPE | 12.54 | 0.99 |

## 4. Conclusions

Introducing low surface tension and high hydrophobicity to polyurethane using copolymerization with fluorinated polyols presents an interesting opportunity to create surface-modified coatings. This work describes the addition polymerization of commercially available perfluoropolyether, which resulted in high molecular weight polymers that retained transparency. The overall aim of this study was to create a fluorinated polyurethane displaying improved surface characteristics using a commercially available fluorinated polyol. Surface segregation of PFPE chain segments was evident in the increasing atomic percentage of fluorine in the PU–*co*–PFPE, with increasing percentages corresponding to increasing PFPE addition. PFPE introduced hydrophobicity to the polyurethane, and increasing amounts of PFPE in polyurethane–*co*–perfluoropolyether improved the hydrophobicity. The critical surface tension of the PFPE–*co*–PU coatings was lower than those of stainless steel and unmodified polyurethane and are comparable to other reported critical surface tensions of modified polyurethanes. Additional characterization in which liquids of a wider range of surface tensions were used to quantify surface energy, and introduction of nanoscale or hierarchical surface topographies, may offer means to further enhance the hydrophobicity of the coatings here. Low-surface-tension polyurethane coatings such as in this study offer a new approach to surface-modified hydrophobic coatings for marine, biomedical, and food production industries. The fluorinated polyurethane polymers produced herein possess promising applications in the food industry as a non-food contact coating to protect and modify equipment surfaces, in addition to other applications in the marine and hospital industries.

**Supplementary Materials:** The following supporting information can be downloaded at: https://www.mdpi.com/article/10.3390/coatings13071133/s1, Figure S1: Peak deconvolution of 1500–900 cm$^{-1}$ for control polyurethane; Figure S2: Peak deconvolution of 1500–900 cm$^{-1}$ for PU–*co*–1PFPE; Figure S3: Peak deconvolution of 1500–900 cm$^{-1}$ for PU–*co*–2PFPE; Figure S4: Peak deconvolution of 1500–900 cm$^{-1}$ for PU–*co*–3PFPE; Figure S5: Peak deconvolution of 1280–1160 cm$^{-1}$ for control polyurethane; Figure S6: Peak deconvolution of 1280–1160 cm$^{-1}$ for PU–*co*–1PFPE; Figure S7: Peak deconvolution of 1280–1160 cm$^{-1}$ for PU–*co*–2PFPE; Figure S8: Peak deconvolution of 1280–1160 cm$^{-1}$ for PU–*co*–3PFPE; Figure S9: Zisman plot of advancing contact angles for four liquids. Second-order polynomial (quadradic) line fitting (solid line). Linear fitting (dotted line); Table S1: Integrations from $^1$H NMR spectra at 3.34–3.5 ppm of perfluoropolyether (A), polyurethane (B), PU–*co*–1PFPE (C), PU–*co*–2PFPE (D), and PU–*co*–3PFPE (E).

**Author Contributions:** Conceptualization, A.M.R. and J.M.G.; Methodology, A.M.R. and J.M.G.; Data curation, A.M.R.; Investigation, A.M.R.; Formal analysis, A.M.R.; Writing—original manuscript writing, A.M.R.; Funding acquisition, J.M.G.; Resources, J.M.G.; Project administration, J.M.G.; Supervision, J.M.G.; Writing—review and editing, J.M.G. All authors have read and agreed to the published version of the manuscript.

**Funding:** This work was supported in part by the United States Department of Agriculture National Institute of Food and Agriculture (Award #2018-67017-27874), Hatch under Accession #1016621, and the Foundation for Food and Agriculture Research (FFAR; Award #CA18-SS-0000000206). This work made use of the Cornell Center for Materials Research Shared Facilities which are supported through the NSF MRSEC program (DMR-1719875).

**Institutional Review Board Statement:** Not applicable.

**Informed Consent Statement:** Not applicable.

**Data Availability Statement:** Available on request.

**Conflicts of Interest:** The authors declare no conflict of interest.

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
