# Peer review of "Synthesis and Characterization of Hydrophobic and Low Surface Tension Polyurethane"

_coatings, doi:10.3390/coatings13071133_

Round 1
Reviewer 1 Report
This paper reports the change in water contact angle achieved by perfluoropolyether-modified polyurethanes, characterizing the processes involved by XPS and infrared spectroscopy. The addition of perfluoropolyether increases the water contact angle of polyurethanes. The water contact angle decreases in small amounts as the amount of perfluoropolyether added to the polyurethane increases. In addition, the addition of perfluoropolyether decreases the surface tension of the material. However, there have been many related works on polyurethanes modified with fluorinated polyethers,The innovation point of this work is insufficient.
1. What is the significance of the research in this paper? When somebody talk about hydrophobicity, the introduction part can briefly mention some hydrophobic related models, applications.
2. Considering the hydrophobic performance, a larger hydrophobic angle can be achieved by designing several aspects that can test the contact angle of more types of liquids on the material surface.
.
Author Response
Comments:
This paper reports the change in water contact angle achieved by perfluoropolyether-modified polyurethanes, characterizing the processes involved by XPS and infrared spectroscopy. The addition of perfluoropolyether increases the water contact angle of polyurethanes. The water contact angle decreases in small amounts as the amount of perfluoropolyether added to the polyurethane increases. In addition, the addition of perfluoropolyether decreases the surface tension of the material. However, there have been many related works on polyurethanes modified with fluorinated polyethers, The innovation point of this work is insufficient.
Response: The authors thank the reviewer for their time and feedback. We agree that the original submission did not sufficiently identify the unique contribution of our work to the field. As noted in the prior response, we have revised our introduction to more clearly define the innovation offered by this manuscript compared to previously published reports.
What is the significance of the research in this paper? When somebody talk about hydrophobicity, the introduction part can briefly mention some hydrophobic related models, applications.
Response: The authors appreciate the feedback. Please see our prior response for notes on research significance, including where we revised the manuscript to address this important point. We have further revised the introduction to include more details about surface energy and hydrophobicity as suggested by the respected reviewer (Lines 59-64).
Considering the hydrophobic performance, a larger hydrophobic angle can be achieved by designing several aspects that can test the contact angle of more types of liquids on the material surface.
Response: We appreciate the comment and fully agree that such design features can enhance hydrophobicity. We have revised our discussion (lines 387-390) to emphasize this important point.
Reviewer 2 Report
The manuscript entitled: ‘Synthesis and Characterization of Hydrophobic and Low Surface Tension Polyurethane’, is interesting and scientifically relevant. The article aims to investigate the selected properties of fluoro-modified polyurethanes synthesized with commercially available fluorinated polyol.
The manuscript raises an important topic, therefore I recommend the work for reconsider after major revision.
Specific comments:
1. Materials (page 2, line: 66-75): in my opinion, this part is too short. It's a very superficial description. These information should be improved about the state of monomers (liquid/ solid/ viscous liquid) and information about average molecular weight, color and melting temperature (if solid monomer). PPG is probably liquid monomer but what about PFPE? Moreover, authors claim that molecular weight of PPG is 725 Da. Molecular weight is not the same what polymerization degree. Scheme 1 B shows PPG with polymerization degree at 725. How much is the average molecular weight of the used PPG? It should be corrected.
2. Table 1, PU synthesis (page 3, line 98) – MDI amount increase with increasing PFPE addition. It seems to be uncorrect, the formulation show that there is not differences in the summary amount of PFPE in the PU-1PFPE, PU-2PFPE and PU-3PFPE…. Am I right?
3. Scheme 1, page 5 – polymerization degree of PPG (B) should be checked; the information about polymerization degree of PFPE (C) should be added or only ‘n’ letter should be added near square brackets; structures D and F - n’ letter should be added near square brackets; moreover fir structures D and F the chain terminated ends on the left sides should be corrected; in this form there is not urethane linkage (COONH) visible but amid (CONH).
4. Results and discussion, Synthesis (page 4, line: 165-168) – Have the syntheses with 4 wt% PFPE been carried out? Were any treatments performed on the molds before the polyurethane casting process? e.g. greased or coated with anti-stick film?
5. Figure 1 (page 6) - ,,….from left to right: polyurethane,….” – there should be given ‘control polyurethane’.
Author Response
The manuscript entitled: ‘Synthesis and Characterization of Hydrophobic and Low Surface Tension Polyurethane’, is interesting and scientifically relevant. The article aims to investigate the selected properties of fluoro-modified polyurethanes synthesized with commercially available fluorinated polyol.
The manuscript raises an important topic, therefore I recommend the work for reconsider after major revision.
Response: The authors gratefully acknowledge the reviewer’s time and effort and appreciate the constructive feedback.
Materials (page 2, line: 66-75): in my opinion, this part is too short. It's a very superficial description. These information should be improved about the state of monomers (liquid/ solid/ viscous liquid) and information about average molecular weight, color and melting temperature (if solid monomer). PPG is probably liquid monomer but what about PFPE? Moreover, authors claim that molecular weight of PPG is 725 Da. Molecular weight is not the same what polymerization degree. Scheme 1 B shows PPG with polymerization degree at 725. How much is the average molecular weight of the used PPG? It should be corrected.
Response: The authors appreciate the reviewers note and agree that we had insufficient detail on the materials used to prepare our polyurethanes. We have added additional information as to the state of the monomers throughout the materials section. In addition, the authors agree with the discrepancy in the molecular weight of PPG and have corrected the reported molecular weight in the materials section and the degree of polymerization in Scheme 1 B to accurately describe the material. We regret the error in our original submission and are grateful to the reviewer for identifying it.
Table 1, PU synthesis (page 3, line 98) – MDI amount increase with increasing PFPE addition. It seems to be uncorrect, the formulation show that there is not differences in the summary amount of PFPE in the PU-1PFPE, PU-2PFPE and PU-3PFPE…. Am I right?
Response: We fully agree that the original manuscript lacked clarity in explaining the differing values of MDI in Table 1. In fact, the numbers in PU-1PFPE, PU-2PFPE and PU-3PFPE reference the wt% of PFPE included in the synthesis, and MDI likewise varies to ensure a 1:1 stoichiometric ratio between MDI and polyols. We have included this detail in the revision to clarify this important point (line 124).
Scheme 1, page 5 – polymerization degree of PPG (B) should be checked; the information about polymerization degree of PFPE (C) should be added or only ‘n’ letter should be added near square brackets; structures D and F - n’ letter should be added near square brackets; moreover fir structures D and F the chain terminated ends on the left sides should be corrected; in this form there is not urethane linkage (COONH) visible but amid (CONH).
Response: The authors regret these errors from the original submission and appreciate the reviewer for identifying them. We have made the noted corrections within Scheme 1.
Results and discussion, Synthesis (page 4, line: 165-168) – Have the syntheses with 4 wt% PFPE been carried out? Were any treatments performed on the molds before the polyurethane casting process? e.g. greased or coated with anti-stick film?
Response: The authors appreciate the comment. Synthesis of 4 wt% PFPE was not carried out during these experiments. The preliminary testing looked at 1 wt%, 3 wt%, and 5 wt% to determine an ideal range for further testing and identified no additional benefit to hydrophobicity, but some degradation of film forming properties, at >3 wt%. Regarding mold pretreatment prior to casting, we have revised the materials and methods to included details on mold preparation details (Lines 140-141).
Figure 1 (page 6) - ,,….from left to right: polyurethane,….” – there should be given ‘control polyurethane’.
Response: The authors appreciate the reviewer’s note and have accordingly revised the figure to include “control polyurethane” in Line 216.**
Reviewer 3 Report
The present work reports the synthesis and characterization of hydrophobic and low surface energy fluorinated polyurethane.
The manuscript is clear, the conclusions straightforward. It merits publication.
A minor comment:
Lines 106-108. Please, report the detector used, as well as the standards used (if a calibration curve was applied for the determination of molecular weights). In addition, I would avoid to report GPC data in Table 2 with such a high accuracy (five significant digits).
Author Response
Comments:
The present work reports the synthesis and characterization of hydrophobic and low surface energy fluorinated polyurethane.
The manuscript is clear, the conclusions straightforward. It merits publication.
Response: The authors thank the reviewer for their time and feedback.
Lines 106-108. Please, report the detector used, as well as the standards used (if a calibration curve was applied for the determination of molecular weights). In addition, I would avoid to report GPC data in Table 2 with such a high accuracy (five significant digits).
Response: The authors appreciate this comment. We have revised our instrument methods to include the detector used (lines 148-150) and revised our reported GPC data to 3 significant digits to more better reflect the sensitivity of the instrument (Table 2).
Reviewer 4 Report
A condensed interesting and scientifically sound paper! I have almost nothing to comment, with one exception. A fast literature search in the Web of Science search tool with the string "polyurethane and perfluoropolyether" returned even 70 results. Expectedly, majority of them are not cited in the Introduction of the submitted paper. Anyway, I suggest that the authors expand a bit the Introduction with citations of some more related papers.
Author Response
Comments:
A condensed interesting and scientifically sound paper! I have almost nothing to comment, with one exception. A fast literature search in the Web of Science search tool with the string "polyurethane and perfluoropolyether" returned even 70 results. Expectedly, majority of them are not cited in the Introduction of the submitted paper. Anyway, I suggest that the authors expand a bit the Introduction with citations of some more related papers.
Response: The authors thank the reviewer for their time and positive feedback. We agree that the original submission did not sufficiently survey the state of the art. We have added additional papers to the discussion and introduction which are directly relevant to the manuscript.
Round 2
Reviewer 2 Report
I recommend the article for publication in present form.
Author Response
Comment:
I recommend the article for publication in present form.
Response: The authors thank the reviewer for their time and positive feedback.